# Cardiac Magnetic Resonance in Patients with Suspected Tachycardia-Induced Cardiomyopathy: The Impact of Late Gadolinium Enhancement and Epicardial Fat Tissue

**DOI:** 10.3390/jpm13101440

**Published:** 2023-09-27

**Authors:** Oleg Orlov, Aref Asfour, Dmitry Shchekochikhin, Zainab Magomedova, Alexandra Bogdanova, Anna Komarova, Maxim Podianov, Grigory Gromyko, Ekaterina Pershina, Alexey Nesterov, Alexandra Shilova, Natalya Ionina, Dennis Andreev

**Affiliations:** 1Department of Cardiology, Functional and Ultrasound Diagnostics, N.V. Sklifosovsky Institute of Clinical Medicine, I. M. Sechenov First Moscow State Medical University, 8 Trubetskaya Str., 119991 Moscow, Russia; viborchik@mail.ru (O.O.); magomedova.zainab.97@mail.ru (Z.M.); pershina86@mail.ru (E.P.);; 2Moscow State Healthcare Institution, City Clinical Hospital №1, 8 Leninsky Ave., 119049 Moscow, Russia; drcor@mail.ru (A.N.); a.s.shilova@gmail.com (A.S.); 3Department Intervention Cardiology and Cardiac Rehabilitation, Pirogov Russian National Research Medical University, 1 Ostrovitianinova Str., 117997 Moscow, Russia; 4Department of Endovascular Diagnostics and Treatment, Russian Biotechnological University (ROSBIOTECH), 33 Talalikhina Str., 109029 Moscow, Russia; 5World-Class Research Center, “Digital Biodesign and Personalized Healthcare”, I. M. Sechenov First Moscow State Medical University, 8 Trubetskaya Str., 119991 Moscow, Russia

**Keywords:** tachycardia-induced cardiomyopathy, cardiac magnetic resonance imaging, late gadolinium enhancement

## Abstract

Tachycardia-induced cardiomyopathy (TIC) is a reversible subtype of dilated cardiomyopathy (DCM) resulting from sustained supraventricular or ventricular tachycardia and diagnosed by the normalization of left ventricular ejection fraction (LVEF) after stable sinus rhythm restoration. The aim of this study was to determine the contribution of cardiac magnetic resonance (CMR) to the differential diagnosis of TIC and DCM with persistent atrial arrythmias in patients hospitalized for the first time with heart failure (HF) with reduced LVEF of nonischemic origin. A total of 29 patients (age: 58.2 ± 16.9 years; males: 65.5%; average EF: 37.0 ± 9.5%) with persistent atrial tachyarrhythmia and first decompensation of HF without known coronary artery diseases were included in this study. The patients successfully underwent cardioversion and were observed for 30 days. The study population was divided into groups of responders (TIC patients; N = 16), which implies achieving FF > 50% or its increase > 10% in 30 days of TIC, and non-responders (N = 13). The increase in left ventricle (LV) volumes measured using CMR was significantly higher in the non-responder group when compared with the responders (114.8 mL ± 25.1 vs. 68.1 mL ± 10.5, respectively, *p* < 0.05). Non-responders also demonstrated decreased interventricular septum thickness (9.1 ± 0.8 vs.11.5 ± 1.3, respectively, *p* < 0.05). Late gadolinium enhancement (LGE) was observed in 12 patients (41.4%). The prevalence of LGE was increased in the non-responder group (25.0% vs. 65.1%, respectively, *p* = 0.046). Notably, a septal mid-wall LGE pattern was found exclusively in the non-responders. Epicardial adipose tissue thickness was decreased in the non-responder group versus the TIC patients. Conclusion: Patients with TIC were found to have smaller atrial and ventricular dimensions in comparison to patients with DCM. In addition, LGE was more common in DCM patients.

## 1. Introduction

### Background

Heart failure (HF) remains one of the main causes of morbidity and mortality in adults and elderly people despite the development of new diagnostic and therapeutic strategies. Tachycardia-induced cardiomyopathy (TIC) is a reversible subtype of nonischemic nonhereditary dilated cardiomyopathy (DCM) resulting from sustained supraventricular or ventricular tachycardia and is implicated in 7–10% of all HF cases with reduced or mildly reduced left ventricular ejection fraction (LVEF). However, the overall prevalence and incidence of TIC are likely to be underestimated [1,2]. It occurs mostly in patients with persistent atrial tachycardia. The main diagnostic feature of TIC is the normalization or increase by 10% of the LVEF after sinus rhythm restoration or rate control achievement. Tachycardia-induced cardiomyopathy refers to the presence of reversible LV dysfunction solely due to an increase in ventricular rate, regardless of the origin of the tachycardia.

One of the most difficult challenges in routine clinical practice is the differential diagnosis of TIC and DCM accompanied by supraventricular tachycardia. Such problems may be caused by the causal relationship between cardiomyopathy and tachyarrhythmia, as its manifestation is initially considered, while the inverse relationship that causes TIC may be overlooked [3,4]. The only established criterion for TIC is the reversibility of myocardial contractility after rhythm or rate normalization. Like other exception diagnoses, TIC should be considered in any patients with reduced myocardial contractility and tachyarrhythmia in the absence of known structural disorders, in particular coronary heart disease (CHD). 

It is generally assumed that TIC develops within a month from the beginning of an episode of sustained tachyarrhythmia in patients with an initially normal heart rate [5]. Nonetheless, cases of acute-onset TIC are described in some reports [6], and animal studies have shown that typical hemodynamic disorders occur within the first 24 h after the onset of tachycardia [7].

Although TIC cases and reports of reversible arrhythmia-related cardiomyopathy have been studied during the last century, to date, only a small number of prospective studies have been conducted. In addition, many research results are limited to small samples of patients who have undergone ablation and are descriptive [1,7,8]. 

In patients with de novo HF and atrial tachycardias, there are several signs to discriminate TIC among them, namely, possible underlying myocardial disease (e.g., myocarditis) and time-dependent contractility improvement on the background of medical treatment. 

Cardiac magnetic resonance imaging (CMR) is one of the most effective methods of heart imaging due to the accurate measurement of heart sizes and volumes, as well as the possibility of quantifying the functional state of the myocardium [9]. The contrast agent used during CMR, gadolinium, has the property of slower accumulation in areas with fibrous changes, for example, in places of scar formation, compared with an unaffected myocardium, which can subsequently be assessed using delayed visualization, known as late gadolinium enhancement (LGE) [10]. Thus, the accurate and reliable assessment of myocardial fibrosis via CMR decreases the need for invasive procedures, such as cardiac biopsies [11,12]. 

Previously published studies have not revealed convincing data on the association between LGE and the recovery of LV contractility in HF in various clinical scenarios, including normalization after cardioversion in patients with DCM and persistent atrial arrhythmias, which allows differential diagnosis with TIC [8,11,13,14,15].

The aim of the study is to determine the impact of LGE presence and other CMR markers on LVEF restoration at 14 and 30 days of follow-up after sinus rhythm conversion in patients with persistent atrial tachycardias and hospitalization for the first onset of HF of nonischemic origin with reduced or mildly reduced EF. 

## 2. Materials and Methods

The present study was conducted from 1 November 2020 to 1 May 2022 in two cardiology departments: the Clinical Center of the I. M. Sechenov First Moscow State Medical University (University Clinical Hospital No. 1), Moscow, Russian Federation, and Moscow State Healthcare Institution “City Clinical Hospital №1 named after N.I. Pirogov, Moscow City Health Department”, Moscow, Russian Federation. The study protocol included a 12-channel resting ECG and transthoracic echocardiography (TTE) upon admission and after direct-current (DC) cardioversion (respondence was estimated according to TTE EF measurement using Simpson’s biplane method), before which transesophageal echocardiography (TEE) was performed to exclude thrombosis of the left atrium auricle. In some cases, pretreatment with amiodarone was used. All patients were treated with DC cardioversion, after which daily ECG monitoring was performed to confirm a stable sinus rhythm before discharge. All initial echocardiographic studies were performed before cardioversion and evaluated by one expert (AB). 

To address the study of patients with possible TIC, the strict exclusion criteria were:Ischemic origin of heart failure: CAD was excluded based on the absence of myocardial infarction in anamnesis or a lack of significant coronary atherosclerosis (>50% stenosis of the left main stem and 70% stenosis in major coronary vessels) according to the results of the coronary angiography (CAG). Patients underwent angiography or CT CAG during the index hospitalization with the exception of patients with known coronary anatomy within the year before hospitalization.Previously known myocardial disease (e.g., hypertrophic cardiomyopathy or toxic cardiomyopathy).Primary valvular heart disease or congenital heart diseases; namely, the exclusion criteria were severe aortic stenosis with a mean gradient ≥40 mmHg, peak aortic velocity ≥4 m/s, and aortic valve area ≤1 cm^2^; severe mitral stenosis; and mitral, tricuspid, and aortic regurgitation of grade 3 according to current guidelines [16].Contraindications for DC cardioversion.Failed DC cardioversion.Patients with left bundle branch block (LBBB).Thyrotoxicosis.Pregnancy or postpartum state.Unwillingness of the patient to participate in the study and to sign the informed consent form.Mental illnesses.Severe concomitant diseases in which life expectancy is not more than a year.

The inclusion criteria were: First hospitalization due to HF;Persistent atrial tachyarrhythmia (>7 days) with heart rate (HR) >100 beats/min;Reduced or moderately reduced LVEF at the time of hospitalization, <50% via echocardiography at admission;No history of previous myocardial infarction and patent coronary arteries on coronary angiography (CAG) or coronary computed tomography angiography.

All diagnostic studies, medical treatment of HF, and the choice in treatment strategy were chosen and carried out according to the direction of the attending physician in accordance with the current guidelines (mostly angiotensin-converting enzyme (ACE) inhibitors, beta blockers, amiodarone, and mineralocorticoid receptor antagonists (MRAs), but angiotensin receptor neprilysin inhibitors (ARNI) and sodium–glucose cotransporter-2 (SGLT2) inhibitors were not available for in-hospital treatment during the study period) [17].

The Charlson comorbidity index was used to measure the burden of concomitant diseases [18]. The index is a validated tool that takes into account the number and the seriousness of main comorbidities.

### 2.1. Cardiac Magnetic Resonance

A cardiac magnetic resonance study was performed during index hospitalization (within 48 h after DC cardioversion or before in the case of pretreatment) with a 1.5T MRI scanner (Vantage ExcelArt TOSHIBA, Tokyo, Japan) using a standardized protocol. All CMR images were analyzed using CMR software CVI 42 (Circle Cardiovascular Imaging, Calgary, AB, Canada) and described by two experts for blind analysis (EP and ZM). Cine images were acquired with a steady-state free-precession pulse sequence in long-axis planes and contiguous 8 mm short-axis slices from the mitral annulus to the apex.

Ten–fifteen minutes after intravenous administration of gadolinium (gadobutrol, Bayer Schering Pharma AG, Berlin, Germany, 0.15 mmol/kg, flow rate of 4 mL/s), delayed-contrast-enhanced images were acquired using inversion recovery fast gradient–echo pulse sequences in the same short-axis locations as the cine images. The inversion time was adjusted (200–300 ms) as needed in the delayed-enhancement image acquisitions to optimally null the signal of the normal myocardium. Chamber quantification and wall motion assessment were performed by the interpreting physician. Representative MR images of responders and non-responders are featured in the Appendix A.

### 2.2. Data Analysis

The presence or absence of LGE was determined by reviewing all short- and long-axis contrast-enhanced images (left ventricular outflow tract, short-axis view from base to apex, long-axis, and four-chamber views). Regions of elevated signal intensity were confirmed in two spatial orientations (Figure 1). The data were calculated by one expert (EP). The amount of fibrosis was assessed manually using the ratio of the mass of fibrosis to the total mass of the myocardium.

### 2.3. CMR Characteristics

The following CMR characteristics were collected for all patients: left ventricular end-diastolic volume index (LVEDV), left ventricular end-systolic volume index (LVESV), LVEF, right ventricular end-diastolic volume index (RVEDV), right ventricular end-systolic volume index (RVESV), left atrium (LA) long and short axis, right atrium (RA) long and short axis, interventricular septal thickness (IVS), and epicardial fat thickness. 

The percentage of LGE was calculated by dividing the LGE mass by the LV mass indexed to the body surface area and then multiplying the quotient by 100. 

The pattern of LGE was stratified through cardiomyopathy: the ischemic pattern was defined as either subendocardial or transmural, and the nonischemic pattern was defined as either subepicardial or mid-wall.

### 2.4. Follow-Up

All patients received the HF treatment described above. All patients underwent TTE at admission and at 14 and 30 days after DC cardioversion. For each patient, the echo follow-up took place at exactly 14 days—apart from two cases when the study was performed on the 15th day due to holidays—and at 30 days. Those who improved their LVEF > 10% or restored > 50% were classified as TIC or responders according to the criteria. The period of 14 days was used due to the available publication data on TIC and the theoretical assumption that the impact on LVEF restoration during the very early phase is more dependent on the TIC mechanism versus the effect of afterload reduction and neurohumoral blockade. The period of 30 days was used for TIC confirmation. All patients with a confirmed TIC diagnosis were scheduled for electrophysiological study and arrhythmia ablation.

### 2.5. Statistical Analysis

All quantitative variables were tested for the presence of a normal distribution using the Kolmogorov–Smirnov test. Variables with a normal distribution were described as an average value with a standard deviation. Variables with a distribution other than normal were described using median and interquartile ranges between the 25th and 75th percentiles and compared using nonparametric tests. To compare the groups using the quantitative variables, Student’s *t*-test (under the condition of a normal distribution) and the Mann–Whitney U test (in the absence of a normal one) were used. Categorical variables are presented in the form of absolute and relative values; for their comparison, depending on the situation, the chi-square criterion or the exact Fisher criterion was used. All tests were 2-sided, and *p* < 0.05 was considered statistically significant. 

A univariate logistic regression analysis was performed to determine the effect of clinical characteristics of patients on the likelihood of recovering EF. All statistical analyses were performed using SPSS software version 22 (SPSS Inc., Chicago, IL, USA).

## 3. Results

The prospective study included 29 consecutive patients (19 males, 65.5%; age, 58.2 ± 16.9 years) with persistent tachyarrhythmia, de novo decompensated HF without known coronary artery diseases. The most frequent comorbidities were arterial hypertension (20 patients, 68.9%) and diabetes mellitus (3 patients, 10.3%). The median of the Charlson comorbidity index was 3 [1; 7] points. 

Three patients (10.3%) had persistent atrioventricular nodal re-entrant tachycardia at admission (AVNRT), two patients (6.9%) had persistent atrial flutter, and the others (twenty-four, 82.8%) had persistent atrial fibrillation. The duration of the arrhythmias was known based on electronic medical records and presented medical documentation. However, in six (20.6%) cases, the duration was based on patients’ self-reported data. Three patients (10.3%) presented to the intensive care unit (ICU) with cardiogenic shock; moreover, one of them required mechanical circulatory support (ECMO) due to hemodynamic instability. In the last case, the patient was presumed to have DCM with atrial fibrillation, and the restoration of contractility and the decrease in LV dimensions after cardioversion was unexpected. The other patients were admitted to the cardiology department. The average EF at admission was 37.0 ± 9.2%, and 10 (34.5%) patients had an EF < 35% (Table 1).

The patients were observed for 30 days after rhythm conversion. Those who experienced an increase in EF > 10% or normalization > 50% were considered responders and were diagnosed with TIC. Sixteen (55.2%) of the patients met the criteria for TIC (responders). Moreover, 15 (51.7%) patients demonstrated restoration of LVEF > 50%.

Responders and non-responders did not differ in age, sex, Charlson comorbidity index, severity of HF or EF, or ongoing therapy at the time of inclusion in the study (*p* > 0.05) (Table 2).

A cardiac magnetic resonance study was performed before cardioversion in nine (31.1%) patients. None of the cases met the criteria for myocarditis (T1-weighted (inflammation, injury) and T2-weighted (edema) sequences, extracellular volume, and LGE) [19]. All studies were readable, and contrast and measured parameters were clearly defined. Interobserver variability for LGE identification between the two physicians was 6.7%. 

Taking into consideration that nine patients underwent CMR before cardioversion, we compared two subgroups of patients who underwent the study before and after DC (Appendix A). Despite the predominance of responders in the group of patients with sinus rhythm, there were no significant differences in age, baseline LVEF, and the distribution of LGE. However, patients who underwent CMR before cardioversion had significantly dilated ventricles and atria. Both groups demonstrated good synchronization and image quality to analyze LGE prevalence and pattern.

CMR data revealed significantly dilated LV in non-responders (114.8 ± 25.1 vs. 68.1 ± 10.5, *p* < 0.05), increased LA long-axis detention in non-responders (3.4 ± 0.5 vs. 2.6 ± 0.2, *p* < 0.05), and increased RA short axis in non-responders (2.8 ± 0.4 vs. 2.3 ± 0.2, *p* = 0.03). Non-responders demonstrated decreased interventricular septum dimension (9.1 ± 0.8 vs.11.5 ± 1.3, *p* < 0.05). 

Late gadolinium enhancement was found in 12 patients (41.4%). The prevalence of LGE was significantly increased in non-responders (25.0% vs. 61.5%, *p* = 0.046) but was also present in 33.3% of complete responders, patients who showed improved EF > 50% in 2 weeks after rhythm restoration. The quantification of LGE resulted in a tendency of a more pronounced amount of LGE in non-responders.

Notably, four patients demonstrated mid-wall septal LGE, a pattern with known negative prognostic impact in several heart diseases [20]. None of the responders were found to have mid-wall septal LGE (*p* = 0.04). 

The thickness of epicardial adipose tissue (EAT) was decreased in non-responders vs. TIC patients (*p* = 0.05).

## 4. Discussion

According to the results of our study, it was revealed that respondents (TIC) had smaller chamber sizes and increased thickness of the CMR in comparison with non-responders. A correlation with EAT was also found. The thickness of the EAT was increased in the TIC patients.

Tachycardia-induced cardiomyopathy has been shown to have several histology features that clearly distinguish it from other forms of nonischemic DCM [21], namely, a moderately increased degree of fibrosis which was lower than those with DCM and inflammatory diseases. Additionally, the presence of macrophages and T-cells was considerably lower in the TIC patients, while the myocardial expression of MHC-2 molecules seemed to be higher. The specific morphology opens the possibility of noninvasive assessment that can be useful in routine clinical practice. 

The use of CMR in patients with HF is becoming more widely used, making it an indispensable part of the disease’s workup and management. There is increasing evidence that CMR data, especially myocardial fibrosis amount, are a risk determinator for sudden cardiac death and implantable cardiac defibrillators’ appropriate use [22,23]. Cardiac magnetic resonance with LGE plays an important role in patients’ workup by assessing for foci of myocardial fibrosis. Moreover, the presence of LGE in patients with nonischemic cardiomyopathy indicates higher mortality and morbidity rates and could indicate worse reversibility of ventricular remodeling [24,25,26,27]. A recent meta-analysis, which included 34 separate studies with a total of 5076 patients (4554 unique patients), concluded that the presence of LGE was associated with a 3.4 [95% CI: 2.04–5.67] higher risk of cardiovascular mortality, a 4.52 risk [95% CI: 3.41–5.99] of ventricular arrhythmic events, and only a 0.15 [95% CI: 0.06–0.36] higher risk of LV reverse remodeling, thus further underscoring the prognostic value of LGE in patients with DCM [23]. 

Despite numerous case reports and case series studies, there are limited data regarding myocardial tissue characteristics in patients with TIC. Addison et al. showed that the presence of LGE was associated with the lack of LVEF recovery in patients with left ventricular systolic dysfunction and atrial fibrillation in a group of 172 patients after undergoing pulmonary vein isolation. The authors inferred that CMR-LGE can be useful in identifying patients who would most benefit from invasive ablative techniques, such as PVI [28]. In a similar study, the CAMERA-MRI (Catheter Ablation Versus Medical Rate Control in Atrial Fibrillation and Systolic Dysfunction) study, a multi-centered prospective randomized clinical trial, enrolled a total of 68 patients with an LVEF < 45% on CMR and a New York Heart Association (NYHA) functional class ≥ II persistent atrial fibrillation without significant coronary artery disease and with no identifiable cause of left ventricular systolic dysfunction. The cohort was consequently equally randomized into two groups, one of which underwent catheter ablation, while the other received pharmacological treatment. All of the patients underwent CMR-LGE. The primary and one of the secondary endpoints were a minimum absolute change in LVEF of 10% between both groups and the effect of the presence and burden of LGE on LVEF improvement. Of interest to our study, the CAMERA-MRI study showed that LGE-positive patients had significantly higher left ventricular end-systolic volume and left ventricular end-diastolic volume when compared to LGE-negative patients (*p* = 0.0114 and *p* = 0.0138, respectively); the lack of ventricular LGE on cardiac MRI imaging was linked to better LVEF improvement and a higher chance of normalization of left ventricular function. However, LGE’s presence did not exclude LVEF restoration; no LGE-negative patients who were treated with catheter ablation fulfilled implantable cardioverter–defibrillator (ICD) implantation criteria at a 6-month follow-up point. In contrast, 21% of the LGE-positive group did meet the specified criteria. The study encouraged the implementation of CMR imaging with LGE as a screening method to prognosticate the reversibility of arrhythmia-induced cardiomyopathy before resorting to invasive treatment methods, such as ablation. For instance, the presence of myocardial scarring with a burden of >10% predicts low reversibility, while its absence or low burden predicts the opposite [17]. Hasdemir et al. studied 298 patients with various types of tachyarrhythmias, of whom 27 had an LVEF < 50%. A group of 22 of these patients experienced an improvement in their LVEF after treatment, and LGE-CMR was performed in 24 of them. The study showed that the absence of LGE was characteristic of TIC; the authors proposed the use of CMR-LGE as a method to differentiate TIC from other primary cardiomyopathies and questioned its possible role in predicting LVEF restoration [11]. Another recent study assessed the use of electrocardiography and CMR imaging in a sum of 43 patients with impaired LVEF and supraventricular tachycardias, 25 of whom were diagnosed with TIC. In contrast to patients who were diagnosed with DCM, those with TIC had narrower QRS complexes, a lower prevalence of LGE, and fewer rehospitalization incidents. The study proposed that LGE is independently associated with the lack of LVEF recovery and worse prognosis [14]. 

Our results found that the presence of LGE increased in patients who did not improve their EF within 2 weeks after sinus rhythm restoration. However, up to 33% of complete responders were found to be LGE-positive. Thus, visualizing LGE in patients with persistent atrial tachycardia and HF with reduced EF decreases the chance of TIC but does not exclude it. Moreover, we observed that the presence of mid-wall septal LGE could be used as a potential marker to differentiate patients with pre-existing cardiomyopathy from TIC patients. 

Epicardial adipose tissue measurement has become an important imaging marker for cardiovascular risk in several clinical settings. We found that the non-responders had decreased thickness compared to the TIC patients. This fact corresponds to a previous study that found decreased EAT in patients with dilated cardiomyopathy compared to controls [29]. A meta-analysis of 3217 individuals (MDCT cardiac scan) from the Framingham Heart Study showed a correlation between EAT volume and the prevalence of AF. This association showed that it is independent of traditional AF risk factors, including BMI [30]. Wong et al. confirmed the relationship between EAT and AF, suggesting that it is stronger than the association between AF and abdominal or overall adiposity [31]. Besides LGE measurement, CMR is the gold standard for ventricle and atrial dimension measurements. We supported data derived from previous TTE-based studies stating that a lower LV volume for the same depressed EF is more common in TIC patients. Jeong et al. stated that a TTE-derived left ventricular end-diastolic dimension less than 61 mm can predict TIC with a sensitivity of 100% and a specificity of 71.4% [32]. Similar results were obtained in another study including 88 patients with newly diagnosed LV systolic dysfunction and tachyarrhythmias. It was shown that a smaller volume and lower LV myocardial mass distinguished patients with TIC from patients with DCM [33]. In the present CMR-derived data, the TIC patients had a significantly lower LVEDV. However, the TTE-derived basal dimensions did not differ between the groups. Thus, we assume that the TIC patients decrease their chamber size according to contractility restoration during short-term follow-up. 

## 5. Limitations

The most significant limitation of our study was the small sample size. Our study is single-centered, and patients with known etiology of HF were excluded. The patient cohort was not unified by only one type of tachyarrhythmia, which might decrease the quality of the results. The duration of arrythmias in several patients was derived from self-reported data, which are more subjective. In addition, MRI was performed both before and after cardioversion, which could affect the comparative assessment of the size of the heart chambers. However, a 30-day follow-up was used to eliminate the impact of the best medical treatment on the results; we acknowledge that different follow-up periods might have led to different results (ex: 14 days vs. 30 days vs. 3 months vs. 6 months). Using advanced CMR, such as T1 mapping, which was unavailable at our institution at the time of the study, could improve myocardial fibrosis identification. 

## 6. Conclusions

Patients with persistent atrial arrhythmia and first hospitalization due to HF with reduced EF improve substantially or restore LV contractility within 14 days after sinus rhythm conversion in 55.2% of cases. The responders (TIC) were found to have smaller chamber dimensions and increased interventricular thickness according to CMR compared to the non-responders. The presence of LGE was increased in the non-responders; however, it was not limited to them. Notably, the septal mid-wall LGE pattern was exclusively found in the non-responders.

## Figures and Tables

**Figure 1 jpm-13-01440-f001:**
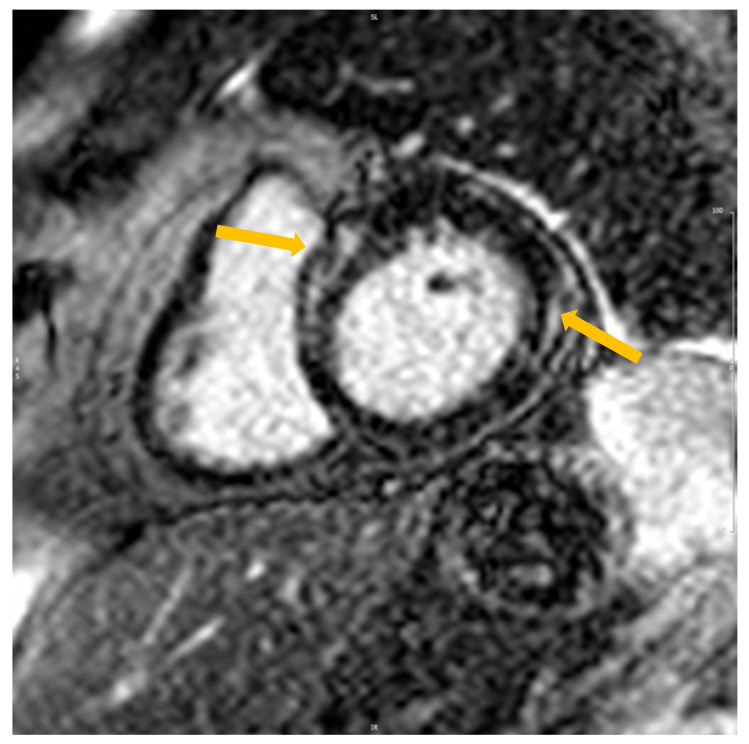
Cardiovascular magnetic imaging with LGE in the short axis in patients fulfilling the criteria of TIC after DC cardioversion. Yellow arrows—the accumulation of LGE.

**Table 1 jpm-13-01440-t001:** Patients’ characteristics.

Patients’ Characteristics (n = 29)	
Age, year	58.2 ± 16.9
Male, n (%)	19 (65.5)
Arterial hypertension, n (%)	20 (68.9)
Diabetes mellitus, n (%)	3 (10.3)
Chronic kidney disease stage 3b, n (%)	5 (17.9)
Charlson comorbidity index, points	3 [1; 7]
Average EF at admission, %	37.0 ± 9.2
Heart rate at admission	120.7 ± 21.0
Atrial fibrillation, n (%)	24 (82.8)
Duration of AFIB before admission, days	40 [11; 360]
Atrial flutter, n (%)	2 (6.9)
Duration of AFL before admission, days	415 [60; 770]
AVNRT, n (%)	3 (10.3)
Duration of AVNRT before admission, days	14 [4; 57]
Cardiogenic shock, n (%)	3 (10.3)
ECMO n (%)	1 (3.4)
Beta blockers before admission, n (%)	5 (17.2)
ACEi before admission, n (%)	8 (27.9)
Digoxin before admission, n (%)	2 (6.9)
Amiodarone before admission, n (%)	6 (20.6)

EF—ejection fraction, AFIB—atrial fibrillation, AFL—atrial flutter, AVNRT—atrioventricular nodal reentry tachycardia, ECMO—extracorporeal membrane oxygenation, and ACEi—angiotensin-converting enzyme inhibitors. Variables with a normal distribution are described as an average value with a standard deviation. Variables with a distribution other than normal are described using median and interquartile ranges between the 25th and 75th percentiles.

**Table 2 jpm-13-01440-t002:** Characteristics of responders and non-responders.

	Responders—TIC (n = 16)	Non-Responders (n = 13)	*p*-Value
Age	62.07 ± 14.40	57.45 ± 14.57	0.43
Male sex, % (n)	58.8% (10)	75% (9)	0.61
Charlson comorbidity index, points	3 [1; 6]	3 [1; 7]	0.9
Duration of persistent arrhythmia, days	52 [1; 721]	32 [2; 246]	0.7
LVEF at admission, %	38.6 ± 10.2	34.8 ± 7.4	1.0
LVEDV baseline TTE, mL	138.8 ± 48.6	118 ± 52.5	0.3
LVEF in 14 days after DC, TTE %	52.6 ± 7.4	41.4 ± 11.5	0.04
RVEDV CMR, mL/m^2^	70.9 ± 13.9	86.3 ± 22.0	0.2
LVEDV CMR, mL/m^2^	68.1 ± 10.5	114.8 ± 25.1	0.04
LA long axis cm/m^2^	2.6 ± 0.2	3.4 ± 0.5	0.04
LA short axis cm/m^2^	2.2 ± 0.2	2.6 ± 0.4	0.07
RA long axis cm/m^2^	2.7 ± 0.3	3.1 ± 0.5	0.09
RA short axis cm/m^2^	2.3 ± 0.2	2.8 ± 0.4	0.03
IVS mm, CMR	11.5 ± 1.3	9.1 ± 0.8	0.04
Patients with LGE, %	25.0	65.1	0.046
% LGE in the whole group	0 [0; 2.65]	2.5 [0; 6.55]	0.07
%LGE among patients with positive LGE	5.2 [3.82; 16.4]	5.7 [2.77; 7.07]	0.6
Septal mid-wall LGE	0	4 (30.7%)	0.04
EAT thickness, cm	4.45 ± 1.16	3.11 ± 0.84	0.03

CMR dimensions were obtained in patients with sinus rhythm at the time of CMR. LVEF—left ventricular ejection fraction. LVEDV—left ventricular end-diastolic volume. RVEDV—right ventricular end-diastolic volume. CMR—cardiovascular magnetic resonance. TTE—transthoracic echocardiogram. DC—direct-current cardioversion. LA—left atrium. RA—right atrium. IVS—interventricular septum. LGE—late gadolinium enhancement. EAT—epicardial adipose tissue. Variables with a normal distribution are described as an average value with a standard deviation. Variables with a distribution other than normal are described using median and interquartile ranges between the 25th and 75th percentiles.

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
