# Peer review of "Cardiac Magnetic Resonance in Patients with Suspected Tachycardia-Induced Cardiomyopathy: The Impact of Late Gadolinium Enhancement and Epicardial Fat Tissue"

_jpm, 2023, doi:10.3390/jpm13101440_

Round 1

Reviewer 1 Report (New Reviewer)

Thank you for the opportunity to review the manuscript entitled “Cardiac magnetic resonance in patients with suspected tachycardia-induced cardiomyopathy”. The differential diagnosis between dilated cardiomyopathy and tachycardiomyopathy (TIC) is one of the most difficult, but also very important, elements in assessing the prognosis of patients with heart failure. One of the first remarks that casts a shadow over this manuscript is the fact that it is written very carelessly, with numerous errors in language, punctuation and typos. Another remark is the very small study group and the fact that the 14-day observation in the case of patients with TIC is too short a period of time. Patients with suspected TIC are observed for several months to determine whether we are really dealing with TIC. In the presented study, the observation period, in my opinion, is a methodological error. This observation period may lead to unrecognized true-responders. Another very important point is that the CMR test is performed with outdated equipment. In the group of patients with AF, there were patients with 1 day of arrhythmia. How was the definition of persistent arrhythmia established for AVnRT - what timing criterion was chosen? How is AVnRT differentiated from AVRT? There are currently a lot of interesting articles about TIC and CMR that have not been included in the discussion.

Due to the above comments, the manuscript is not suitable for publication. Unfortunately, it is burdened with methodological errors that only cause confusion. From a clinical point of view, the use of data from this article may lead to diagnostic errors.

Manuscript is written very carelessly, with numerous errors in language, punctuation and typos. 

Author Response

Dear editorial board of Personalized Medicine Journal!

I am writing to submit our revision of the manuscript entitled “Cardiac magnetic resonance in patients with suspected tachy-cardia-induced cardiomyopathy: impact of late gadolinium enhancement and epicardial fat tissue” for consideration for publication in the Journal

We thank you for reviewing our manuscript and for your comments.

When editing our manuscript, we took into account the outlined comments.

The answers to the several comments are listed below

«Thank you for the opportunity to review the manuscript entitled “Cardiac magnetic resonance in patients with suspected tachycardia-induced cardiomyopathy”. The differential diagnosis between dilated cardiomyopathy and tachycardiomyopathy (TIC) is one of the most difficult, but also very important, elements in assessing the prognosis of patients with heart failure.

One of the first remarks that casts a shadow over this manuscript is the fact that it is written very carelessly, with numerous errors in language, punctuation and typos.

We have carried out a proofreading of the text.

Another remark is the very small study group and the fact that the 14-day observation in the case of patients with TIC is too short a period of time. Patients with suspected TIC are observed for several months to determine whether we are really dealing with TIC. In the presented study, the observation period, in my opinion, is a methodological error. This observation period may lead to unrecognized true-responders.

We evaluated the data on the 30th day of observation, however, no significant changes were found in our sample.

Another very important point is that the CMR test is performed with outdated equipment. In the group of patients with AF, there were patients with 1 day of arrhythmia. How was the definition of persistent arrhythmia established for AVnRT - what timing criterion was chosen?

There was a disappointing misspell in the first version of the article. Instead of "1" it should be "11"

 How is AVnRT differentiated from AVRT?

AVNRT was proved with electrophysiological study which was carried out after 1-3 months (added to the text)

There are currently a lot of interesting articles about TIC and CMR that have not been included in the discussion.

The data has been updated

Due to the above comments, the manuscript is not suitable for publication. Unfortunately, it is burdened with methodological errors that only cause confusion. From a clinical point of view, the use of data from this article may lead to diagnostic errors.»

We have tried to rework and supplement our work. We hope for a positive decision

Reviewer 2 Report (New Reviewer)

In general, the authors have performed an interesting study about a miscellaneous topic of heart failure. Nevertheless, there is a need for major corrections. My comments for your work are the following:

1. In the affiliations please mention the authors' (academic) titles as well as the names of their institutions.

2. Please, correct grammatic and syntax errors (i.e. "impact" of CMR-in the abstract, "medwall" etc..) throughout manuscript.

3. The style of references is not uniform through the reference list. Please, use a program in order to display references according to the journal's recommendations (i.e. EndNote, Zotero, ...).

4. In the introduction, you should present the definition of tachycardia-induced cardiomyopathy according to latest guidelines (1st paragraph). Also, presentation of data needs rearrangement (i.e. tachycardia-induced cardiomyopathy diagnosis/differential diagnosis and limitations in one paragraph-then, present information about CMR in another paragraph). Moreover, the aim of your study needs to be presented in a more precise/cohesive way.

5. Please, provide the full name of any abbreviation in the first place of its appearance and then use the abbreviated term throughout the manuscript (i.e. DC, PVI,...).

6. In the methods, please provide a separate paragraph about ethics and informed consent procedure.

7. In the methods please provide the appropriate references for each of your evaluations (i.e CMR evaluation, follow-up etc). Also, define the criteria for responders/non-responders to treatment and provide the appropriate references.

8. In the statistical analysis please clarify that continuous variables with normal distribution were presented as mean and standard deviation.

9. In the results (and throughout the whole manuscript) please explain what do you mean by the term "coronary artery diseases".

10. In the results, you could also present information about other risk factors of your study population such as smoking status and dyslipidemia since they might have affect your results as well as of exercise habits. Also, there is no need to present reference 20 in the section of results (but in the discussion). Moreover, improve the presentation of the tables (provide table captions and explanations about the types of your variables i.e. continuous, normality of distribution etc.).

11. Relatively to responders and non-responders, you could perform an analysis according to the presence of arrhythmias (i.e. atrial fibrillation, atrial flutter, AVNTR) as a comparator variable.

12. Discussion needs extensive improvement. The first paragraph should be omitted and replaced by the major findings of your study. Then, in the following paragraphs you have to discuss each of your findings according to the published knowledge of the literature and make comparisons. Moreover, you could enrich the data about epicardial fat thickness.

The manuscript needs extensive English grammar/syntax corrections (i.e. you could use an appropriate service).

Author Response

Dear editorial board of Personalized Medicine Journal!

I am writing to submit our revision of the manuscript entitled “Cardiac magnetic resonance in patients with suspected tachy-cardia-induced cardiomyopathy: impact of late gadolinium enhancement and epicardial fat tissue” for consideration for publication in the Journal

We thank you for reviewing our manuscript and for your comments.

When editing our manuscript, we took into account the outlined comments.

«The answers to the several comments are listed below. In general, the authors have performed an interesting study about a miscellaneous topic of heart failure. Nevertheless, there is a need for major corrections. My comments for your work are the following:

  1. In the affiliations please mention the authors' (academic) titles as well as the names of their institutions.

Fixed

  1. Please, correct grammatic and syntax errors (i.e. "impact" of CMR-in the abstract, "medwall" etc..) throughout manuscript.

Fixed

  1. The style of references is not uniform through the reference list. Please, use a program in order to display references according to the journal's recommendations (i.e. EndNote, Zotero, ...).

- References fixed

  1. In the introduction, you should present the definition of tachycardia-induced cardiomyopathy according to latest guidelines (1st paragraph). Also, presentation of data needs rearrangement (i.e. tachycardia-induced cardiomyopathy diagnosis/differential diagnosis and limitations in one paragraph-then, present information about CMR in another paragraph). Moreover, the aim of your study needs to be presented in a more precise/cohesive way.

- The European clinical guidelines do not specify an exact definition. Provided a link to the definition

  1. Please, provide the full name of any abbreviation in the first place of its appearance and then use the abbreviated term throughout the manuscript (i.e. DC, PVI,...).

– Done

  1. In the methods, please provide a separate paragraph about ethics and informed consent procedure.

- Done

  1. In the methods please provide the appropriate references for each of your evaluations (i.e CMR evaluation, follow-up etc). Also, define the criteria for responders/non-responders to treatment and provide the appropriate references.

- Defined and provided for 14 and 30 days follow-up

  1. In the statistical analysis please clarify that continuous variables with normal distribution were presented as mean and standard deviation.

Yes, added to the manuscript

  1. In the results (and throughout the whole manuscript) please explain what do you mean by the term "coronary artery diseases".

The patients had no significant structural pathology according to coronary angiography

  1. In the results, you could also present information about other risk factors of your study population such as smoking status and dyslipidemia since they might have affect your results as well as of exercise habits.

In order not to overload the tables with information, the data focus was shifted towards the data associated with HF and CMR. The total number of smokers was 6 (21%), patients with dyslipidemia were 12 (41%). There was no significant difference between the groups. The absence of coronary atherosclerosis, indicating a low risk, was one of the inclusion criteria

Also, there is no need to present reference 20 in the section of results (but in the discussion). Moreover, improve the presentation of the tables (provide table captions and explanations about the types of your variables i.e. continuous, normality of distribution etc.).

Done

  1. Relatively to responders and non-responders, you could perform an analysis according to the presence of arrhythmias (i.e. atrial fibrillation, atrial flutter, AVNTR) as a comparator variable.

We tried to remove the AVNRT. At the same time, with a decrease in the sample, the significance of LGE for the non-responders and TIC groups decreased (53.3% and 45.5%, respectively). At the same time, the LGE pattern remained significance (33.3 vs. 0, p = 0.03). EAT has lost significance due to a small sample while maintaining the trend p=0.08

  1. Discussion needs extensive improvement. The first paragraph should be omitted and replaced by the major findings of your study. Then, in the following paragraphs you have to discuss each of your findings according to the published knowledge of the literature and make comparisons. Moreover, you could enrich the data about epicardial fat thickness.»

Added

Reviewer 3 Report (New Reviewer)

Oleg et al have submitted a manuscript titled, “Cardiac magnetic resonance in patients with suspected tachy- cardia-induced cardiomyopathy: impact of late gadolinium enhancement and epicardial fat tissue”, which shows that patients who experience an improvement or recovery of EF with control of their tachyarrhythmia have less LGE and smaller cardiac chamber volumes, compared to patients who do not.

The major issue in my opinion, is that the timing of cardiac MRI was not consistent across the group, before or after cardioversion. As noted by the authors, the cardiac chamber sizes vary before and after cardioversion.

I would also recommend consistency in using terms of responders or tachycardiac-induced cardiomyopathy and non-responders or dilated cardiomyopathy. In some instances, it is unclear whether the authors are referring to a broad population of dilated cardiomyopathy patients, or to the patients who did not respond after cardioversion. I would also caution against assuming the non-responders are dilated cardiomyopathy unrelated to the tachyarrhythmia. Certainly, some proportion may have HFrEF with a coexisting tachyarrhythmia that was not the cause of the reduced EF, however, it is also plausible that these tachycardia-induced cardiomyopathy patients presented at a later stage in their disease process, which is also supported by larger chamber sizes and more fibrosis. As we know, patients with HFrEF of > 1 year duration are less likely to experience improvement in dimensions and EF with guideline-directed medical therapy. The same could apply to treatment of arrhythmia as well.

If available, adding data about HF medications and BNP values would add to the robustness of the study.

Suggest including MRI images of a representative responder and non-responder patient.

To assess response, are the pre and post EF’s derived from echo or cardiac MRI or a combination? If this is an echo derived EF, is this from visual estimation, Teicholz 4 chamber method or 3D? There should be consistency in this assessment.

Were functional severe MR and TR also excluded, or only primary, severe MR?

Minor edits:  

Exclusion criteria should be moved up and mentioned just below inclusion criteria.
Change tachysystole to tachycardia

Minor grammatical errors 

Author Response

Dear editorial board of the Journal of Personalized Medicine!

We are writing to submit our revision of the manuscript entitled “Cardiac magnetic resonance in patients  with suspected tachycardia-induced cardiomyopathy: impact of late gadolinium enhancement and epicardial fat tissue.” for consideration for publication in the Journal

We thank you for reviewing our manuscript and for your comments.

When editing our manuscript, we took into account the outlined comments.

The answers to the several comments are listed below

«The major issue in my opinion, is that the timing of cardiac chamber sizes vary before and after cardioversion»

The time interval of MRI after cardioversion was no more than 48 hours (added to the text). We compared the structural parameters of the heart in patients who performed MRI before or after cardioversion and, indeed, the size of the right atrium in patients with sinus rhythm was shorter, which is reflected in the supplementary table and limitations (added). At the same time, the timing of MRI does not affect the presence of LTE in the myocardium, so it does not affect the main conclusions.

«I would also recommend consistency in using terms of responders or tachycardiac-induced cardiomyopathy and non-responders or dilated cardiomyopathy. In some instances, it is unclear whether the authors are referring to a broad population of dilated cardiomyopathy patients, or to the patients who did not respond after cardioversion. »

The distribution of patients into groups in our study was based on the initial identification of responders. Accordingly, the detection of a significant increase in EF after the tachyarrhythmia abortion is the criterion for TIC. In this study, we are not referring to the general population of DCM, but only to the population of DCM + tachycardia.

«I would also caution against assuming the non-responders are dilated cardiomyopathy unrelated to the tachyarrhythmia. Certainly, some proportion may have HFrEF with a coexisting tachyarrhythmia that was not the cause of the reduced EF, however, it is also plausible that these tachycardia-induced presented at a later stage in their disease process, which is also supported by larger chamber sizes and more fibrosis. As we know, patients with HFrEF of >1 year duration are less likely to experience improvement in dimensions and EF with guideline-guided medical therapy. The same could apply to treatment of arrhythmia as well. »

We do not claim that non-responders are dilated cardiomyopathy unrelated to the tachyarrhythmia. In fact, within the framework of the study, non-responders are equivalent to non-TIC. The associations of DCM and tachycardia are multifaceted, we are only partially expanding the field of knowledge regarding the relationship of TIC and myocardial fibrosis.

«If available, adding data about HF medications and BNP values would add to the robustness of the study. »

At the time of the study, the measurement of BNP and list of previously used medications were not included in the local clinical protocol.

«Suggest including MRI images of a representative responder and non-responder»

Added to supplementary data.

«To assess response, are the pre and post EF’s derived from echo or cardiac MRI or a combination? If there is an echo derived EF, is this from visual estimation, Teicholz 4 chamber method or 3D? There should be consistency in this assessment. »

Respondence was estimated according echo derived EF measurement by Simpson’s biplane method, added to the Materials and Methods section.

«Were functional severe MR and TR also excluded, or only severe MR?»

In paragraph 3 of the exclusion criteria were mentioned, that all severe valvular diseases were excluded.

«Minor edits: Exclusion criteria should be moved up and mentioned just below inclusion criteria. Change tachysystole to tachycardia»

Done

This manuscript is a resubmission of an earlier submission. The following is a list of the peer review reports and author responses from that submission.

Round 1

Reviewer 1 Report

The Authors present the study entitled "Cardiac magnetic resonance in patients with suspected tachycardia-induced cardiomyopathy”. I should admit that the idea of that research is of great interest and could be important from the clinical point of view. However, some essential comments were crucial in my final decision.

Major comments:

1. It is unclear what is the real novelty of the study. The Introduction part needs a thorough background, and the Discussion part needs to address the obtained results to literature data. There is a lot of discordances between the presented by the authors results and data from the literature; however, the authors did not explain that accordingly.

2. How could the Authors use their results in clinical practice? What could be the real applicability of the study? Some notes are to be in the discussion part.

3. There are some weak points in the methodology of the study (how the authors could diverse between reduced ejection fraction due to tachyarrhythmia and tachyarrhythmia due to initial heart failure?; the information in lines 77- 84 is not compatible with the exclusion criteria; how the authors calculated the duration of arrhythmia?; LA size should be presented as LAVI).

4. The small sample size was probably the main reason for the lack of significant differences. For that sample, comparing median and quartiles (Q25 – Q75) would be the better statistics option for all comparisons.

Minor comments:

1. Many sentences in the introduction need to be cited by data from the literature.

2. The text could be more coherent and easier to read.

3. What the information in Table 2, “LGE among patients with positive LGE”, means?

Author Response

Thank you for  thoughtful review. We  address each comment in attached  document and  it the  main body of  the  paper.

Reviewer 2 Report

The paper is interesting, but some improvements are needed.

Lines 13-14: its should be clarified that the main inclusion criteria of EF  were detected by Echocardiography (report reference for methodology). It should be also reported in detail if the first echocardiographic examination, to diagnose TIC, was performed after cardioversion. The 14 days of follow-up are really a short time and this should be discussed in further details in the limitations (recovery of LV function could be observed even later).  The extact time of echocardiographic follow-up for the patients should be reported as mean +- range in days. Is it possible that exactly 14 days was the follow-up time for each one of all the patients? Blind Inter and intra observer variability of echocardiographic measurements should be reported. 

Lines 55-57: please report some references. 

Line 63: duration of tachyarrhytmias should be also reported, possibly for each types of arrhythmias as reported in Table 1. It should be stated if only persistent atrial fibrillation, atrial Flutter and AVNRT were considered. In line 147 AVNRT; in Table 1, line 168: AVRT (please check).

Abbreviations should be reported and explained in the legenda of Table 1 and 2.  

Line 64: report Ref. for classification of EF < 50%: McDonagh TA, Metra M, Adamo M, Gardner RS, Baumbach A, Böhm M, Burri H, Butler J, ÄŒelutkien , Chioncel O, Cleland JGF,Coats AJS, Crespo-Leiro MG,Farmakis D, Gilard M,Heymans S, Hoes AW,Jaarsma T, Jankowska EA,Lainscak M, Lam CSP, Lyon AR, McMurray JJV, Mebazaa A, Mindham R, Muneretto C, Francesco Piepoli M, Price S, Rosano GMC, Ruschitzka F, Kathrine Skibelund A; ESC Scientific Document Group. 2021 ESC Guidelines for the diagnosis and treatment of acute and chronic heart failure.  Eur Heart J. 2021 Sep 21;42 (36):3599-3726. doi: 10.1093/eurheartj/ehab368. 

Line 65: "patent"

Line 76: check english: .. "with life expectancy is..."

Line 77: please give more detail on the cut-off of severity of valvular heart disease and congenital heart disease to exclude the patients from the present analysis. By using echocardiography mild valvular heart disease or regurgitation are very frequently present, but those subjects have not necessarily to be excluded. References should be indicated. 

Line 81: define "significant".

Lines 85,86: please report treatment after cardioversion. 

Line 86: report Ref for guidelines. 

Line 87: Charlson Comorbidity Index: please describe in some detail and report a Reference. 

Lines 89-90: its is not clear how the informations given by CMR were used. In 9 cases CMR was performed before cardioversion. In these cases heart rate was probably very high: could this have influenced time resolution, ECG synchronization and measured parameters, and even perhaps the LGE assessment? If so, these patients should be excluded. Was myocardial edema assessed by CMR and how this could have modified LGE evaluation during the acute phase? How myocarditis was excluded?

Line 99: report References.

Line 100: "interpreting"

Lies 104-105: blind inter and intraobserver variability of CMR measurements  and LGE assessment should be reported. 

Figure 1: put arrows to indicate LGE. 

Line 121: Current treatment at discharge should be reported in detail. 

Line 128: Statistical analysis: why ANOVA within and between groups was not used? 

Line 139: "statistical". 

Line 142 and 144: avoid repetitions. 

Line 147: "nodal".

Try to put together Table 1 and Table 2 with 3 columns, one for the totality, one for responders and one for non responders.

Table 3 is very difficult to understand and confusing. In the 9 cases with AF heart rate was probably very high: could this have influenced CMR time resolution, ECG synchronization and measured parameters, and even perhaps the LGE assessment? If so, all these patienst shoudl be excluded. Furthermore it is not clear which parameters reported in Table 3 are derived by CMR and which by Echocardiography. 

Line 222-223 and Table 4: could be this due to the great heterogeneity of these patients with so many variables and bias? Please discuss in detail. 

Line 253: "LGE-CMR"

Line 271: "fibrillation"

Line 306: "tachyarrhythmias"

Lines 314-323: Limitations should be better before Conclusions. 

Line 340: "involved"

Line 422, 424, 425: please check typing errors.

Line 452: "gadolinium"

Line 457: "Cardiomyopathy".

Please check spelling throughout the paper. 

Author Response

We appreciate for  reviewing our  paper.  Our team responded on most comments  in attached  document

Round 2

Reviewer 1 Report

The main idea of the study could be of great clinical interest. However, the study is not ready for publication due to many weak points I mentioned in my previous revision. The Authors did not perform sufficient changes in the study; therefore, I could not change my previous decision.

Reviewer 2 Report

In 9 cases CMR was performed before cardioversion. In these cases heart rate was probably high and should be reported as mean heart rate and ranges. This should be compared with the mean heart rate and ranges for the other CMR cases performed after cardioversion. It should also be clearly stated if this have influenced time resolution, ECG synchronization and measured parameters, and even perhaps the LGE assessment. 

Blind Inter and intra observer variability of CMR measuremtes should be obtained, separating the information in the 9 cases before cardioversion and all the others.  

Minor editing required.